# State of the Art in Robotic Surgery with Hugo RAS System: Feasibility, Safety and Clinical Applications

**DOI:** 10.3390/jpm13081233

**Published:** 2023-08-06

**Authors:** Francesco Prata, Alberto Ragusa, Claudia Tempesta, Andrea Iannuzzi, Francesco Tedesco, Loris Cacciatore, Gianluigi Raso, Angelo Civitella, Piergiorgio Tuzzolo, Pasquale Callè, Matteo Pira, Matteo Pino, Marco Ricci, Marco Fantozzi, Salvatore M. Prata, Umberto Anceschi, Giuseppe Simone, Roberto M. Scarpa, Rocco Papalia

**Affiliations:** 1Department of Urology, Fondazione Policlinico Universitario Campus Bio-Medico, 00128 Rome, Italy; alberto.ragusa@unicampus.it (A.R.); andrea.iannuzzi@unicampus.it (A.I.); francesco.tedesco@unicampus.it (F.T.); loris.cacciatore@unicampus.it (L.C.); gianluigi.raso@unicampus.it (G.R.); a.civitella@policlinicocampus.it (A.C.); p.tuzzolo@policlinicocampus.it (P.T.); pasquale.calle@unicampus.it (P.C.); matteo.pira@unicampus.it (M.P.); matteo.pino@unicampus.it (M.P.); ma.ricci@unicampus.it (M.R.); marco.fantozzi@unicampus.it (M.F.); r.scarpa@policlinicocampus.it (R.M.S.); rocco.papalia@policlinicocampus.it (R.P.); 2Department of General Surgery, Fondazione Policlinico Universitario Campus Bio-Medico, 00128 Rome, Italy; claudia.tempesta@unicampus.it; 3Simple Operating Unit of Lower Urinary Tract Surgery, SS. Trinità Hospital, Sora, 03039 Frosinone, Italy; mario.prata@libero.it; 4Department of Urology, IRCCS “Regina Elena” National Cancer Institute, 00144 Rome, Italy; umberto.anceschi@gmail.com (U.A.); puldet@gmail.com (G.S.)

**Keywords:** Hugo RAS, Medtronic, minimally invasive, review, state of the art

## Abstract

Since its introduction in the early 2000s, robotic surgery has represented a significative innovation within a minimally invasive surgery approach. A variety of robotic platforms have been made available throughout the years, and the outcomes related to those platforms have been described in the literature for many types of surgeries. Medtronic’s Hugo^TM^ RAS system is one of the newest robotic generations launched, but because of its recent placing on the field, comprehensive clinical data are still lacking. The aim of the present state of the art is to address the current literature concerning the use of the Hugo^TM^ RAS robot in order to report its feasibility, safety and clinical applications in different surgical branches. Two reviewers independently conducted a search on the “PubMed” electronic database, using the keywords “Hugo” and “Hugo RAS”. After the initial screening of 35 results, a total of 15 articles concerning the Hugo RAS system were selected for the review, including both oncological and benign surgery. Patients’ demographic and baseline data were compared including, when available, docking system times, complications and oncological outcomes in the fields of urologic, gynecologic and general surgery. With reference to urological procedures, a total of 156 robot-assisted radical prostatectomies, 10 robot-assisted partial nephrectomies, and 5 robot-assisted adrenalectomies were performed, involving a total of 171 patients. The surgical branch in which the Hugo system found its major application was urology, which was followed by gynecology and general surgery. The Hugo RAS system by Medtronic represents an innovative and safe surgical platform, with excellent perspective for the future and different clinical applications in many surgical branches. More studies are needed to validate the safety and results from this new robotic platform.

## 1. Introduction

Since the early 2000s, in the context of a minimally invasive surgical approach, robot-assisted surgery (RAS) has represented a significative innovation, and a variety of robotic platforms were described throughout the years for different specialties.

After the DaVinci platform, many robotic systems have been introduced to the market to overcome the intrinsic limitations of the Intuitive^®^ System. CMR in Cambridge (UK) has developed a platform known as Versius [1]. It consists of a modular system provided with an open console and high-definition 3D glasses. Additionally, there are three or four separate bedside arm-carts units, providing the surgeon with a more customized approach to surgical planning. Notably, unlike the DaVinci console, the Versius system’s controls are entirely hand-controlled. 

On this background, the Hugo^TM^ RAS System (Medtronic, Minneapolis USA©) was developed as a modular robotic platform featuring four independent arm-carts, enabling the adaptation of surgical strategies for highly customized procedures [2]. The introduction of the Hugo^TM^ RAS System aimed to provide an alternative robotic platform that offers a more ergonomic and personalized working environment. Among its notable technical advantages are a more ergonomic trocar position, a larger working space for the bedside assistant, and cost-effectiveness for individual procedures. It comprises a console, a system tower, and four independent robotic arm-carts. The system is designed to enable surgeons to perform complex procedures with greater precision and control while reducing patient trauma, pain and recovery time, offering several key features. These include superior dexterity and range of motion, precise instrument control, high-definition three-dimensional (3D) visualization, haptic feedback, and remote telemetry. The system also includes advanced safety features such as collision avoidance, force sensing, and automatic joint lockout. In addition, the Hugo^TM^ RAS modular asset can be configured to support various surgical specialties such as urology, gynecology, colorectal, and general surgery.

In the urological field, in which the system has been primarily used, Hugo^TM^ RAS was employed for radical and simple prostatectomies, radical cystectomies, radical and partial nephrectomies, and pyeloplasty procedures [3,4,5,6,7,8,9,10,11,12]. In gynecology, the system can be used for hysterectomies, myomectomies, and sacrocolpopexies [13,14]. For colorectal surgery, the system can be employed for colectomies, rectopexies, and low anterior resection procedures [15]. In general surgery, the system has been used for cholecystectomies, appendectomies, and hernia repair surgeries [16,17,18].

Hugo^TM^ RAS is currently being used in different hospitals in Asia which have found its main field of application in general surgery procedures, such as cholecystectomies, inguinal hernia repairs, lower anterior resections and gastric bypasses and in urologic surgeries such as prostatectomies. In 2021, the latter was the very first procedure carried out with Medtronic’s Hugo^TM^ RAS System in the Asia Pacific Region by Dr. Ragavan [11]. Furthermore, the recent regulatory approval of the robotic system by Japan, the third largest robotic market worldwide, represents an outstanding expansion of minimally invasive surgeries in Asia, thus making the platform more accessible to a larger number of surgeons.

Hugo^TM^ RAS offers several benefits to both patients and surgeons. On the patient’s side, the system has been shown to reduce postoperative pain, blood loss, and hospital stay, while also improving cosmetic outcomes and overall quality-of-life. For surgeons, the system offers greater control, precision, and dexterity, which can lead to improved outcomes, reduced surgical errors, and enhanced ergonomics. In addition, thanks to its modularity, this robotic platform can potentially help surgeons in performing complex procedures with ease and confidence. Due to his recent employing in the surgical scenario, comprehensive clinical data are still lacking.

In this “state of the art”, we aimed to resume the current literature regarding the Hugo^TM^ RAS System experience in order to report its feasibility, safety and clinical applications in different surgical branches.

## 2. Research and Literature

A review of original articles was performed using PubMed/Medline and Scopus in May 2023. Two reviewers independently conducted the research using the keywords “Hugo”, and “Hugo RAS”. After an initial screening of 35 results from Pubmed, a total of 15 articles concerning the Hugo RAS system were selected for review, including both oncological and benign surgery (Figure 1). 

Manuscripts not written in the English language and not focused on the Hugo^TM^ RAS system were excluded. Reviews, conference/meeting abstracts and editorial comments were also excluded. No time window related to the articles was defined. Full-text articles were assessed individually and data were extracted and then cross-checked between authors in order to have a double check of the methodological quality of the data extraction itself. Patients’ demographic and baseline data were compared. When available, docking system times, complications, and oncological outcomes in the fields of urologic, gynecologic and general surgery were collected. Being the aim of this review narrative due to the novelty of the robotic platform as the lack of extensive literature about the system, it was not possible to fulfil all the PRISMA criteria so as to register it with the PROSPERO database. The goal of this review was to provide a state-of-art as to summarize the relevant features of the Hugo^TM^ RAS System across different surgical specialties, and to come up with a foundation for discussion for surgeon to pursue future opportunities and applications of this robotic platform.

## 3. Results

### 3.1. Urological Procedures

#### 3.1.1. Robot-Assisted Radical Prostatectomy

##### Robot-Assisted Radical Prostatectomy with the Novel Hugo Robotic System: Initial Experience and Optimal Surgical Setup at a Tertiary Referral Robotic Center


The first case series of five robot-assisted radical prostatectomies (RARPs) ± lymph node dissection performed with the new Hugo^TM^ RAS system in Europe was described by C.A. Bravi et al. [10]. The study was conducted at Onze Lieve Vrouwziekenhuis Hospital (Aalst, Belgium). There was no need for conversion or for the placement of new ports; all procedures were carried out safely. No intraoperative complications or system technical issues were noted. The independent arm-carts represent an important advantage where a lot of surgical configurations can be performed based on the preferences of the surgeon or patient peculiarities. In this five-patient case series, the bed-assistant surgeon was located at the right of the patient while the scrub nurse worked on the left side of the patient with easier access to the left robotic arms. Docking time was not reported, but a step-by-step docking procedure is encouraged in order to reduce the estimated time. 

The median console time was 120 min, and the median operating time was 170 min (interquartile range (IQR): 140–180). The average length of stay was 3 days (range, 2–4). For a surgical team that had received the necessary training, starting the system and docking the robotic arms represented simple and quick procedures.

##### Implementation and Outcomes of Hugo^TM^ RAS System in Robotic-Assisted Radical Prostatectomy


C. G. Alfano et al. [9] retrospectively analyzed 15 consecutive patients who underwent RARP with the Hugo^TM^ RAS System from June to October 2021 in this robotic experience. A trans-peritoneal approach was performed in a lithotomy position, using four robotic trocars and two laparoscopic ports for the assistant. The safety and clinical feasibility of this platform were assessed through satisfactory perioperative outcomes. Moreover, no conversions or major complications occurred. 

The first docking occurred in approximatively 15 min due to the setup of the carts, amounting to the longest time noted. Then, in the following procedures, a median time of 7 min docking per case was reached. 

Median operative time was 235 min (IQR: 213–271), which is comparable with other robotic platforms, and median estimated blood loss was 300 mL (IQR: 100–310). Positive surgical margins were reported in 5 patients (33%). The median hospitalization time was 2 days (2-2), and the median time to remove the foley catheter was 7 days (7-7). Four weeks after surgery, all patients had undetectable PSA values, and 61% of them were continent. 

##### Outcomes of Robot-Assisted Radical Prostatectomy with the Hugo RAS Surgical System: Initial Experience at a High-Volume Robotic Center


In this article, C. A. Bravi et al. [8] retrospectively described 112 consecutive patients who underwent RARP ± extended pelvic lymph-node dissection (ePLND) at OLV Hospital (Aalst, Belgium) between February and November 2022. The median age was 65 years (IQR, 60–70) and the median preoperative prostate-specific antigen (PSA) was 7.9 ng/mL (5.8–10.7). An International Society of Urological Pathology (ISUP) grade group 3 tumor on prostate biopsy was reported in 34% patients. On preoperative magnetic resonance imaging (MRI), a suspect of extraprostatic disease was noted in 26 (23%) patients. Docking time was not reported. The median operative time was 180 min (IQR 145–200), and 27 (24%) men underwent ePLND. This is the first report regarding urinary continence (UC) recovery and final surgical pathology for RARP performed through the Hugo^TM^ RAS system with a sample size greater than 100 patients.

Thirty-four (31%) had extraprostatic extension of the disease, and 10 (9%) had positive surgical margins at final pathology. Regarding PSA after surgery, 88% (60/68) had undetectable PSA (<0.1 ng/mL). 

The odds of UC recovery was 36% (95% confidence interval (CI), 28–47%) at 1 month and 81% (95% CI, 72–89%) at 3 months. The median time to UC recovery was 36 days (95% CI, 34–44). 

##### Robot-Assisted Laparoscopic Radical Prostatectomy Utilizing Hugo RAS Platform: Initial Experience


In this article, Ragavan et al. [7] described an initial experience of patients who underwent RARP performed with Hugo^TM^ RAS at the Apollo Hospitals (Chennai, Tamil Nadu, India). In addition, the authors provided a comparison of the outcomes with a similar series of RARP carried out through the DaVinci robotic system during a similar period in a non-randomized study fashion.

A total of 34 patients were included in two groups. In detail, 17 (50%) radical prostatectomies were achieved with the Hugo^TM^ RAS system, and the other 17 were performed with the DaVinci system. 

The total operative time (210 vs. 195 min) and docking time (19 vs. 17 min) were similar between the two groups. A radical resection and vesicourethral anastomosis were achieved in all cases. No major intra- or postoperative complications in up to 1-month follow-up were detected.

##### The New Surgical Robotic Platform HUGO^TM^ RAS: System Description and Docking Settings for Robot-Assisted Radical Prostatectomy


In this paper, Totaro et al. [2] provided a description of the system setup and surgical approach in an initial experience of RARP performed with the Hugo^TM^ RAS system. The study has been conducted at Fondazione Policlinico Universitario Agostino Gemelli IRCCS, Università Cattolica del Sacro Cuore, Rome, Italy.

After an official training, seven consecutive patients affected by localized prostate cancer underwent RARP with the HUGO^TM^ RAS system. 

Port placement has shown to be safe, effective, and easy replicable. A total of six trocars were positioned, out of which four were robotic and two were laparoscopic trocars. Robotic ports included a 12 mm optical trocar located 1 cm above the umbilicus, and three more 8 mm trocars for the other independent carts. The assistant is positioned on the right of the patient, with an easy access to the 5 mm and 12 mm laparoscopic trocars. All ports were inserted straight along a transversal line set at 14 cm from the upper limit of the pubic bone, except for the 5 mm assistant trocar that was placed on the midline between the endoscope trocar and the first right lateral robotic trocar, about 5 cm up toward the head of the patient. Eight centimeters were observed between all the ports.

Operative times appeared to be easily reproducible and comparable to those obtained with the DaVinci system. No major system faults and conflicts between robotic arms were observed after the first procedure. 

Some tips and tricks were provided after this experience. The principal clashing problem was identified between carts 2 and 3. It should be avoided, keeping these two not too close to each other, with a maximum separation between ports 2 and 3. All the instruments should be always under direct visualization of the surgeon, who should avoid abrupt movements during the procedure.

##### Robot-Assisted Radical Prostatectomy Feasibility and Setting with the Hugo^TM^ Robot-Assisted Surgery System


In 2022, Sarchi et al. [6] tried to assess the RARP feasibility and to describe their step-by-step technique for the operative setup with the Hugo^TM^ RAS system on a preclinical model. Three RARPs on cadavers were performed by a single experienced surgeon. Trocar configuration was displayed as following: one 11 mm endoscope port was placed on the midline above the umbilicus, 16–18 cm from the pubic bone; another three 8 mm robotic ports were placed under vision on a transversal line 5 cm below the optics, with at least 8 cm between each other. Finally, two assistant ports were positioned in the right hemi-abdomen.

The endoscope arm-cart was the first docked, then left and right robotic arms. 

In their setting, docking and tilt angles were:Endoscope: 175°; minus 45°;Surgeon left hand 140°; minus 30°;Surgeon right hand 225°; minus 30°;Fourth arm 105°; plus 30°.

Three procedures on male cadavers were performed successfully with a trans-peritoneal approach following objectively validated steps: bladder detachment, endopelvic fascia incision, bladder neck dissection, dissection of vasa and seminal vesicles, dissection of posterior space between the prostate and rectum, lateral dissection of the prostate, suture of the dorsal venous complex, apical dissection, posterior reconstruction, and vescico–urethral anastomosis. The mean docking and operative times were 5 and 90 min, respectively, and they were comparable to their clinical experience with other robotic systems. No intraoperative complications were recorded, and there was no need for conversion or additional ports placement. No clashing occurred between the robotic arms and the bed-side assistant. Their setup did not lead to any intraoperative complication, clash of the instruments, or system failure. In conclusion, the authors reported the first experience with the Hugo^TM^ RAS System with RARP and showed that this novel platform performed well and safely, as all procedures were carried out without any technical failures. Notwithstanding the satisfactory outcomes, the preclinical nature of this study precluded confirming the feasibility and safety of Hugo^TM^ RAS for radical prostatectomy, and further studies in a clinical setting are needed to confirm the safety in terms of intraoperative complications, absence of system failure, and instruments clashes.

#### 3.1.2. Robot-Assisted Partial Nephrectomy

##### Feasibility and Optimal Setting of Robot-Assisted Partial Nephrectomy with the Novel “Hugo” Robotic System: A Preclinical Study


In this preclinical study conducted by Bravi et al. [5], a transperitoneal-approached partial nephrectomy (PN) was carried out on three male cadavers (two left PN and one right PN). The mean total operative time was 105 min (out of which 7 min were of mean docking time and 85 min were of mean console time). Complications, described as the damage of abdominal organs, did not occur. No robotic arm clashing was observed. 

##### Initial Experience of Robot-Assisted Partial Nephrectomy with Hugo^TM^ RAS System: Implications for Surgical Setting


Gallioli et al. [4] aimed to describe the setting and report the performance of the first series of RAPN performed with the Hugo^TM^ RAS system. 

Ten consecutive patients who underwent four four-arm configuration transperitoneal RAPNs at Fundació Puigvert, Autonomous University of Barcelona between February and December 2022 were prospectively enrolled. The right and left patient side were seven and three, respectively. The median tumor size and PADUA scores were 3 (2.2–3.7) cm and 9 (8–9), respectively. The median docking and console time were 9.5 (9–14) and 138 (124–162) minutes, respectively. The median warm ischemia time was 13 (10–14) minutes, and one case was performed clamp-less. The median estimated blood loss was 90 (75–100) mL. One major complication (Clavien-Dindo 3a) occurred. No case of positive surgical margin was recorded. 

A full flank position with a 60° angle between the patient and the bed was adopted. 

An 11 mm endoscope trocar was positioned on the mid-clavicular line, 5 cm under the edge of the ribs, in the instance of left RAPN. Then, keeping 8 cm from the endoscope trocar, two 8 mm robotic trocars were positioned on the pararectal line. The fourth arm’s 8 mm trocar is positioned 8 cm apart from the right-hand trocar, 2 cm above the mid-clavicular line. The assistant’s 12 mm trocar was positioned beneath the endoscope trocar. The space between trocars and bony prominences was kept at 2 cm. The first arm to be linked was the cranial cart, which was positioned behind the patient’s head with a 30° tilted-down posture and a 45° docking angle. The endoscopic cart was then positioned close to the patient, tilted down 30 degrees and docked at a 90-degree angle. The fourth arm cart was positioned below the patient’s legs at a 135° docking angle and a 30° tilt-down position. Finally, the surgeon’s right-hand cart was positioned in front of the patient’s legs at a 215° docking angle and the maximum tilt-up. Each cart was positioned 45–60 cm away from the operating table.

In the instance of right RAPN, a specular trocar placement technique was used, keeping 2 cm between robotic trocars and bony prominences and 8 cm between robotic trocars. To retract the liver, a second 5 mm assistance port was cranially attached to the right arm trocar. The carts positioning scheme was specular.

One case of conversion to laparoscopic partial nephrectomy was reported, where the combination of a suboptimal trocars’ placement and hepatomegaly caused continuous clashing between robotic arms.

#### 3.1.3. Robot-Assisted Adrenalectomy

##### The New Robotic Platform Hugo^TM^ RAS for Lateral Transabdominal Adrenalectomy: A First World Report of a Series of Five Cases


In the study report of Raffaelli et al. [18], five patients with benign adrenal pathology were treated with lateral transabdominal total adrenalectomy. Expert surgeons had to complete a specific training at ORSI academy, delivered by Medtronic, before scheduling the surgeries. Three 73-year-old, 65-year-old and 78-year-old females (patient n.1, 2 and 3, respectively) with Cushing syndrome were enrolled in the study, along with a 30-year-old female with a left para adrenal tumor and a 57-year-old male with left pheochromocytoma (patient n. 4 and 5 respectively). No intraoperative complications occurred, but a total of five arm collision instances were observed (three for patient n.1, one for patient n.2 and one for patient n.4). A single postoperative complication happened in patient n.4, for whom the dissection of the para adrenal cyst attached to the tail of pancreas caused a postoperative increase in amylase values that lasted for three days and was treated with a few days of fasting. The authors believe that the challenging dissection could demonstrate Hugo’s capabilities rather than fallacies, since no conversion was needed. The total operative time was of 99, 139, 85, 153, and 119 min in chronological order, while the docking and console time were 8, 6, 5, 5 and 5 min and 54, 55, 29, 108 and 61 min, respectively. The hospital stay lasted two days for every patient except for the one in which the complication occurred, whose hospital stay was 8 days.

#### 3.1.4. Robot-Assisted Laparoscopic Surgeries for Nononcological Urologic Disease: Initial Experience with Hugo RAS System

The present retrospective observational study from Elorrieta et al. [3] presents good clinical outcomes in the robot-assisted pyeloplastic treatment of a 70-year-old female patient with a CT documented left pyeloureteral junction stenosis, presenting with acute pyelonephritis and 48% left renal function on MAG3 scan. The total operative time was 201 min, of which 108 min were console time and 11 min were docking time. A double J stent was inserted as the tutor, along with a drainage and urethral catheter, which were removed after 6 weeks, 1 week and 2 days post-intervention. The postoperative course was uncomplicated. In the same study, the following clinical cases are described: a 60-year-old female with a left ureteral stone of 2.5 × 1 × 1 cm and preoperative creatinine levels of 1.17 mg/dL treated with RAS laparoscopic ureterolithotomy. The total operative time was 150 min, out of which 110 and 9 min were the console and docking time, respectively. A double J stent was positioned and removed 4 weeks post-surgery. No complications were reported; a 46-year-old male with left distal ureteral stenosis underwent RAS laparoscopic ureteral reimplant. Clinical course was uneventful. The surgical time took 226 min (8.6 min for docking and 164 min at console). The left double J stent, the drain and urethral catheter were removed after 6 weeks, 3 days and 1 week; a 32-year-old female with right distal ureteral stenosis had a RAS laparoscopic ureteral reimplant carried out. The surgical procedure took 257 min, while the docking and console time took up 5.7 and 119 min, respectively. Once again, postoperative evolution was uncomplicated. The drain was removed after two days, the urethral catheter was removed after one week, and the double J stent was removed after 6 weeks; a 43-year-old male with left renal atrophy was successfully treated with RAS laparoscopic nephrectomy that lasted for 222 min (console time 95 min, docking 8.1 min) and discharged two days after surgery.

### 3.2. General Surgery Procedures

The current literature about the use of Hugo RAS is mainly focused on two specific surgical branches: urology and gynecology. There are only three reports in which Hugo RAS is implemented in general surgery, but even if the underlying pathologies are very different, each article points out the safety and feasibility of the robotic platform in the single case scenario. More clinical data are necessary in order to define and possibly standardize each procedure later described. 

#### 3.2.1. First Worldwide Report on Hugo RAS^TM^ Surgical Platform in Right and Left Colectomy 

This technical note by Bianchi et al. [15] aims to report the first three hemicolectomies performed with the Hugo RAS platform and to describe the related operating room setup, robotic arm placement and trocar layout. Two female patients 66 and 74 years old underwent right colectomies (RC), and one 75-year-old male was treated with left colectomy (LC). The complete mesocolic excision and high vascular ligation were carried out in all surgeries. Patients were preoperatively diagnosed with colonic adenocarcinoma (site: cecum, transverse colon and sigmoid colon, respectively). A single surgical team composed by colorectal surgeons experienced in robotic surgery had to complete a specific surgical training program at ORSI academy. The three procedures were completed with no intraoperative or postoperative complications. Clinical outcomes were favorable. The mean docking time, of 8 min, was similar to the ones reported with other robotic systems, while the longer total operative times (of 336 and 365 min for RC and 340 m for LC) were presumably due to the still scarce experience with Hugo RAS. The recorded console times were of 255, 265, and 360 min, respectively. The mean length of postoperative stay was 5 days. No system failure was recorded, although few adjustments had to be made on one right colectomy because of arm conflict; the modifications did not interfere with the total operative time. This is one of the very first articles that describes the use of Hugo RAS in general surgery; thus, more clinical data are necessary in order to standardize Hugo-RAS robotic colectomies. It also calls for the need to implement and refine the ergonomics of surgical devices and robotic arms. 

#### 3.2.2. Robot-Assisted Nissen Fundoplication with the New HUGO^TM^ Robotic-Assisted System: First Worldwide Report with System Description, Docking Settings and Video

The present case report, by Quijano et al. [17], is the first in which the possible advantages related to the known major flexibility of Hugo-RAS independent robotic arms are applied to treat a patient affected by hiatal hernia. The patient, a 65-year-old female with no significant medical history except for a GERD associated to the hiatal hernia, underwent robotic Niessen fundoplication. While the intraoperative surgical steps were comparable to the ones in the literature, the article describes the specific docking setup developed by Medtronic for this surgery. Robotic Niessen surgery with Hugo RAS is feasible and safe, with no complications reported. The total operative and docking times (of 100 min and 3 min respectively) were similar to the ones obtained with other robotic platforms, although the authors hope that those times could decrease with more experience. The hospital stay lasted three days. The modular concept of Hugo RAS, and its open console setting, are once again highlighted as factors that could increase the surgeon’s comfort; possible disadvantages of this conformation rely on the need to individually position each robotic arm. Coherently to the report of Bianchi et al., this article suggests that the introduction of devices already in use in laparoscopic surgery, such as LigaSure and Harmonic scalpel, could improve performance.

#### 3.2.3. Initial Experience in a Novel Robotic System: Hugo^TM^ Robotic Assistant Surgery System for Robotic Inguinal Hernia Repair

The possibility to use only three out of four robotic arms, in general surgery, finds one of its best applications in abdominal wall reconstruction surgery, as described in the case report of Balachandran et al. [16]. A robotic trans-abdominal pre-peritoneal repair was carried out in a 56-year-old male with bilateral inguinal hernia. The clinical operative and postoperative course was uncomplicated. The patient was discharged the following day. No recurrence was reported on a 3-month follow-up. After a brief description of the surgical steps, the authors expose the main features found in Hugo that could possibly be related to a better surgical performance compared to the Da Vinci system, including the previously cited individual cart arms and open design surgeon console, as well as the different conformation of the operating rig—which is trigger-like—that increases the ergonomics compared to other platforms’ grips. While the use of Hugo RAS on abdominal wall defects was shown to be safe, the scarcity of the current literature imposes further research in order to define its role in hernia surgery repair.

### 3.3. Gynecological Surgery

#### 3.3.1. The First European Gynecological Procedure with the New Surgical Robot Hugo RAS. A Total Hysterectomy and Salpingo-Oophorectomy in a Woman Affected by BRCA-1 Mutation

The case report by Monterossi et al. describes the first European gynecological procedure with Hugo RAS, consisting of a total hysterectomy with bilateral salpingo-oophorectomy [13]. A 62-year-old woman with no significant medical history but affected by BRCA1 mutation underwent a prophylactic total extra-fascial hysterectomy with bilateral salpingo-oophorectomy. The docking time was 6 min and total operative time was 58 min. No intra- or postoperative complications nor robotic arm clashing were reported. Blood loss was negligible (<30 mL). The drain was removed on the second postoperative day. The pain VAS score progressively decreased after surgery, and the patient was discharged on the second postoperative day. Following an explanation of system setup and trocar placement, the article highlights how the possible advantages related to the system conformation do not necessarily imply a greater efficiency. The latter could improve with an accurate preoperative spatial study aimed to avoid trocar and cart arm collisions. 

#### 3.3.2. Robotic Sacrocolpopexy Plus Ventral Rectopexy as Combined Treatment for Multicompartment Pelvic Organ Prolapse Using the New Hugo RAS System

In the case report by Campagna et al. [14], a 68-year-old woman with multicompartmental prolapse, rectocele, bladder neck obstruction and obstructed defecation syndrome underwent nerve sparing RSCP plus ventral rectopexy. The procedure was carried out following a standardized technique described by the same surgical group. The docking time was 8 min, while the console time was 120 min; the total operative time was 165 min. The patient was discharged 2 days after surgery. The clinical course was uneventful, and follow-up at 3 months was satisfactory. 

Table 1 summarizes our findings and the main data from the studies included.

## 4. Discussion

Currently, the Hugo^TM^ RAS system represents the most captivating robotic platform available due to its improved modularity, representing an important advantage for surgical configurations that can be performed based on the surgeon preferences or on patient peculiarities. Since the Hugo^TM^ RAS system has independent arm-carts, a step-by-step docking procedure is encouraged in order to reduce docking time [5,6,10]. At a first impression, the open console and new design of the hand controls could be faced as a challenge to the learning curve due to years of experience in a different platform with another operative setting. However, once the robot is docked and the instruments are placed, the high-definition 3D image provided by the 3D glasses did not change the approach to the surgery. In addition, by using extra glasses, other surgeons and visitors around the console can see the same operative 3D image as the surgeon. Moreover, the pistol-like controllers and settings did not interfere in the surgical technique, although requiring an adaptation period until the mastering of the different buttons to lock and unlock the arms was completed. During consecutive steps of RARP, we believe that the instruments provided appropriate traction and dissection capacity without delaying or interfering with the intraoperative performance. 

However, the docking process is more challenging and demands training because all arms are attached to individual carts that must be placed in the correct position with an appropriate arm angulation. If these parameters are not respected, the optimal angles and arm movements will be compromised during the surgery, thus possibly resulting in different complications such as robotic arm clashing [4]. 

In addition, during critical surgical steps as suturing or performing anastomosis, an useful feature of the system can be used: it is the possibility to increase the scaling factor for wrist rotation, facilitating these challenging steps. Interestingly, wrist rotation can be electronically enhanced via a multiplier (up to ×2), with a rotation range of 520°. Last but not least, the use of an intelligent system for pressure-guided insufflation is advisable during the early experience, which allows the maintenance of a stable pneumo-peritoneum and may contribute to reduce the operative time. 

With regard to the surgeon console, the ‘‘pistol-grip’’ handles allow the movement of instruments with the thumb and index finger, while the pistol trigger represents the clutch function. Finally, the open surgical console improves team communication and allows multiple observers (e.g., trainees) to follow the operation using 3D vision. 

The feasibility of the Hugo^TM^ RAS system has been investigated in various studies and clinical trials [5,6]. It has demonstrated promising results in terms of technical feasibility and its ability to be integrated into surgical workflows. Surgeons have reported the system to be intuitive, ergonomic, and easy to use, which facilitates the adoption of robotic-assisted procedures. The system’s advanced instruments and features, such as 3D high-definition visualization and wristed instruments with seven degrees of freedom, offered enhanced dexterity and precision during surgical procedures.

Safety is of paramount importance in any surgical system, and the Hugo^TM^ RAS system has undergone rigorous testing and evaluation to ensure patient safety. The system incorporates multiple safety mechanisms, including collision avoidance technology, intelligent instrument tracking, and advanced imaging modalities, which enable surgeons to perform procedures with enhanced accuracy and reduced risk. Additionally, the system provides haptic feedback to the surgeon, enabling them to detect and avoid tissue damage during surgery.

The Hugo^TM^ RAS system has been primarily used in urological procedures such as RARPs and RAPNs [3,4,5,8,9,10,11]. These procedures involve the removal of the prostate or a portion of the kidney, respectively. The system’s precise movements, stable camera platform, and enhanced visualization aid surgeons in performing these complex procedures with improved results. Clinical studies have reported reduced blood loss, shorter hospital stays, faster recovery times, and improved functional and oncological outcomes compared to traditional open surgeries.

About future perspectives, Hugo^TM^ RAS is still a relatively new technology, and its future seems to be promising. The system’s modularity and versatility allow supporting the development of new surgical techniques and procedures. In addition, continued advancements in robotics, imaging, and artificial intelligence are likely to further improve the system’s capabilities and increase its potential applications. Research is also ongoing to evaluate the system’s long-term outcomes, safety, and cost-effectiveness.

The review of the literature focused on the use of the Hugo^TM^ RAS robot in both oncological and benign surgery has highlighted promising results. Overall, the platform is considered feasible and safe compared to other robotic and non-robotic systems, although further evidence is needed in order to evaluate long-term surgical outcomes and the follow-up of oncological patients. Few studies also suggest that the introduction of Hugo^TM^ RAS could reduce the costs of robotic-related surgeries, thus possibly extending its field of clinical applications.

Despite its numerous advantages, Hugo^TM^ RAS also has some limitations. The system requires extensive training and skills to operate. It also has longer setup and preparation times compared to traditional laparoscopic surgery, which may limit its use in some settings. Moreover, the system’s robotic arms can be bulky and may require additional port sites, which can increase the risk of complications such as bleeding, infection, and tissue trauma.

## 5. Conclusions

Hugo^TM^ RAS by Medtronic is an advanced surgical system that has the potential to transform the field of minimally invasive surgery. Its key features, applications, benefits, limitations, and future perspectives make it an attractive option for many surgical specialties. However, more research is still needed to fully understand the system’s safety, efficacy, and cost-effectiveness. As the technology continues to evolve, it is likely to become an increasingly important tool for surgeons seeking to provide the best possible outcomes for their patients while operating in the most comfortable setting.

## Figures and Tables

**Figure 1 jpm-13-01233-f001:**
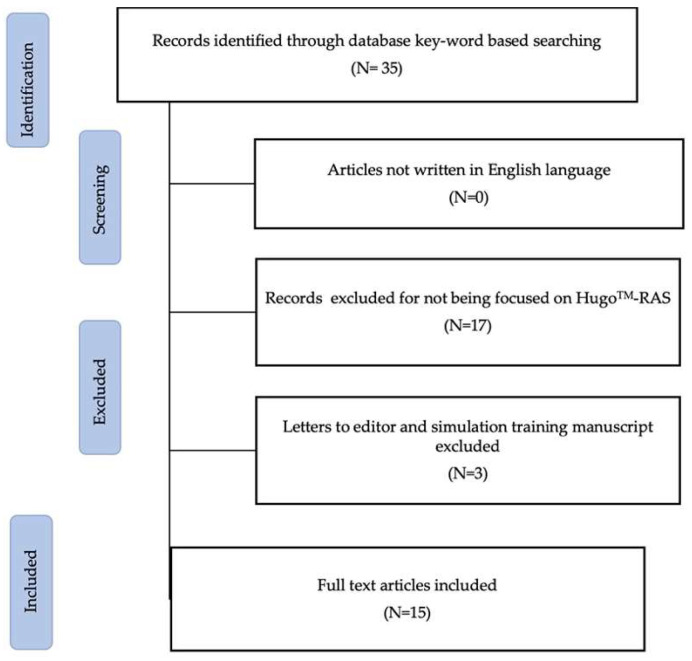
Graphic study design showing the flowchart of study selection.

**Table 1 jpm-13-01233-t001:** Characteristics and data of studies included divided for specialty.

Author	Title	Year	Specialty	Number of Patients and Type of Study	Intervention	Docking Time	Operative Time	Outcome
C.A. Bravi et al. [10]	Robot-Assisted Radical Prostatectomy with the Novel Hugo Robotic System: Initial Experience and Optimal Surgical Setup at a Tertiary Referral Robotic Center	2022	Urology	5 patients Case Series	RARP	Not reported	170 min	After receiving the necessary training, starting the system and docking the robotic arms were simple and quick procedures.
C. G. Alfano et al. [9]	Implementation and Outcomes of Hugo(^TM^) RAS System in Robotic-Assisted Radical Prostatectomy	2023	Urology	15 patients Case Series	RARP	7 min	235 min	Four weeks after surgery, all patients had undetectable PSA values, and 61% of them were continent.
C. A. Bravi et al. [8]	Outcomes of Robot-Assisted Radical Prostatectomy with the Hugo RAS Surgical System: Initial Experience at a High-Volume Robotic Center	2023	Urology	112 patients Case Series	RARP ± ePNLD	Not reported	180 min	The odds of UC recovery was 36% (95% confidence interval [CI], 28–47%) at 1 month and 81% (95% CI, 72–89%) at 3 months. The median time to UC recovery was 36 days (95% CI, 34–44).
N. Ragavan et al. [7]	Robot-Assisted Laparoscopic Radical Prostatectomy Utilizing Hugo RAS Platform: Initial Experience	2023	Urology	34 patients Comparison with DaVinci	RARP	19 min	210 min	A radical resection and vesicourethral anastomosis were achieved in all cases. No major intra- or postoperative complications in up to 1-month follow-up were detected.
A. Totaro et al. [2]	The New Surgical Robotic Platform HUGO^TM^ RAS: System Description and Docking Settings for Robot-Assisted Radical Prostatectomy	2022	Urology	7 patients Case Series	RARP	Not reported	Not reported	Operative times appeared to be easy reproducible and comparable to those obtained with the DaVinci system. No major system faults and conflicts between robotic arms were observed after the first procedure.
Sarchi L. et al. [6]	Robot-Assisted Radical Prostatectomy Feasibility and Setting with the HugoTM Robot-Assisted Surgery System	2022	Urology	3 cadavers Preclinical study	RARP	5 min	90 min	Docking and operative time were comparable to their clinical experience with other robotic systems. No intraoperative complications were recorded, no need for conversion or additional ports placement. No clashing occurred.
C. Bravi et al. [5]	Feasibility and Optimal Setting of Robot-Assisted Partial Nephrectomy with the Novel “Hugo” Robotic System: A Pre-Clinical Study	2022	Urology	3 cadavers Preclinical study	RAPN	7 min	105 min	Complications, described as damage of abdominal organs, did not occur. No robotic arm clashing was observed.
A. Gallioli et al. [4]	Initial experience of robot-assisted partial nephrectomy with Hugo^TM^ RAS system: implications for surgical setting	2023	Urology	10 patients Case Series	RAPN	9.5 min	138 min	One case of conversion to laparoscopic partial nephrectomy was reported, where the combination of a suboptimal trocars’ placement and hepatomegaly caused continuous clashing between robotic arms.
M. Raffaelli et al. [18]	The new robotic platform Hugo^TM^ RAS for lateral transabdominal adrenalectomy: a first world report of a series of five cases	2023	Urology	5 patients Case Series	Lateral trans-abdominal total adrenalectomy	8, 6, 5, 5 and 5 min	99, 139, 85, 153, and 119 min	No intraoperative complications occurred, but a total of 5 arm collision instances were observed (three for patient n.1, one for patient n.2 and one for patient n.4). A single postoperative complication happened.
V. Elorrieta et al. [3]	Robot assisted laparoscopic surgeries for nononcological urologic disease: initial experience with Hugo RAS system	2023	Urology	4 patients Retrospective Observational Study	Robot-assisted pyeloplasty, ureterolithotomy, ureteral reimplant, and nephrectomy	11 min 9 min 5.7 min 8.1 min	201 min 150 min 257 min 222 min	Comparable results to the previous robotic systems, suggesting the multiple potential uses of the Hugo RAS.
P. Bianchi et al. [15]	First worldwide report on Hugo RAS^TM^ surgical platform in right and left colectomy	2023	General Surgery	3 patients Case Series	Robot-assisted colectomy	8 min (median)	336, 340, 365 min	No system failure was recorded, although few adjustments had to be made on 1 right colectomy because of arm conflict; the modifications did not interfere with the total operative time.
Y. Quijano et al. [17]	Robot-assisted Nissen fundoplication with the new HUGO^TM^ Robotic assisted system: first worldwide report with system description, docking settings and video	2023	General Surgery	1 patient Case Report	Robot-assisted Nissen fundoplication	3 min	100 min	Showed safety and feasibility of Nissen fundoplication for hiatal hernia with the Hugo™ RAS system and provided relevant data that may assist early adopters of this surgical platform.
P. Balachandran et al. [16]	Initial experience in a novel robotic system: Hugo^TM^ robotic assistant surgery system for robotic inguinal hernia repair	2022	General Surgery	1 patient Case Report	Robot-assisted inguinal hernia repair	Not reported	Not reported	Clinical operative and postoperative course was uncomplicated. Patient was discharged the following day. No recurrence was reported on a 3-month follow-up.
G. Monterossi [13]	The first European gynecological procedure with the new surgical robot Hugo RAS. A total hysterectomy and salpingo-oophorectomy in a woman affected by BRCA-1 mutation	2022	Gynecology	1 patient Case Report	Robot-assisted total hysterectomy and salpingo-oophorectomy	6 min	58 min	Gynecological surgery with Hugo™ RAS seems feasible, safe and effective as shown by initial experiences in urological surgery.
G. Campagna et al. [14]	Robotic sacrocolpopexy plus ventral rectopexy as combined treatment for multicompartment pelvic organ prolapse using the new Hugo RAS system	2023	Gynecology	1 patient Case Report	Robot-assisted sacrocolpopexy	8 min	120 min	Patient was discharged 2 days after surgery. Clinical course was uneventful, and follow-up at 3 months was satisfactory.

## Data Availability

The data presented in this study are available on request from the corresponding author.

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
