# Peer review of "State of the Art in Robotic Surgery with Hugo RAS System: Feasibility, Safety and Clinical Applications"

_jpm, 2023, doi:10.3390/jpm13081233_

Round 1

Reviewer 1 Report

This review is a well described study that evaluated the feasbility and safety of new robotic system in different medical fields. The methodology is ok. the discussion is good.

Author Response

We appreciate the positive feedback and the time to carefully review the manuscript.

Reviewer 2 Report

The authors reviewed the application of the new surgical robotic platform HUGOTM RAS System in multiple surgeries, including urology, gynaecology, colectomy, and general surgery. The HUGOTM RAS System offers many benefits for both patients and surgeons. This review paper provided a comprehensive introduction to its feasibility, safety and clinical applications in different surgical branches. Here are some suggestions that the authors should be addressed carefully.

1. An introduction to robotic-assisted surgery should be included in the background. What types of surgical robotic-assisted systems are used in clinical? Compared to other systems, what are the advantages and disadvantages of the HUGOTM RAS System?

2. Does any  Asia centre or hospital use the System? Any causes?

3. A graphic study design is needed. 

4. Tables can be used to demonstrate the applications among urology, gynaecology, colectomy, and general surgery.

The paper writing should be improved a lot.

For example, Line 87-89, Corrected sentence: The first case series of five robot-assisted radical prostatectomies (RARPs) ± lymph node dissection performed with the new HugoTM RAS system in Europe was described by C. A. Bravi et al [1].

Lines 105-106, Corrected sentence: C. G. Alfano et al [2] retrospectively analyzed 15 consecutive patients who underwent RARP with the HugoTM RAS System from June to October 2021 in this robotic experience.

Author Response

The authors reviewed the application of the new surgical robotic platform HUGOTM RAS System in multiple surgeries, including urology, gynaecology, colectomy, and general surgery. The HUGOTM RAS System offers many benefits for both patients and surgeons. This review paper provided a comprehensive introduction to its feasibility, safety and clinical applications in different surgical branches. Here are some suggestions that the authors should be addressed carefully.

We thank the reviewer for the comment, the positive feedback, and the time to carefully review the manuscript and to give incisive yet constructive comments, which has greatly helped us improve this revised draft. Our point-by-point response to each specific comment is below, as follows:

  1. An introduction to robotic-assisted surgery should be included in the background. What types of surgical robotic-assisted systems are used in clinical? Compared to other systems, what are the advantages and disadvantages of the HUGOTM RAS System?

The manuscript was improved accordingly and an introduction to robot-assisted surgery as well about types of robotic systems was added. Moreover, advantages and disadvantages of the HUGOTM RAS System have been described. The modifications were highlighted in the manuscript as follow:

After DaVinci platform, many robotic systems have been introduced to the market to overcome intrinsic limitations of the Intuitive® System. CMR in Cambridge (UK) has developed a platform known as Versius. It consists of a modular system provided with an open console and high-definition 3D glasses. Additionally, there are three or four separate bedside arm-carts units, providing the surgeon with a more customized approach to surgical planning. Notably, unlike the DaVinci console, the Versius system's controls are entirely hand-controlled.

On this background, the HugoTM RAS System (Medtronic, USA©) was developed as a modular robotic platform featuring four independent arm-carts, enabling the adaptation of surgical strategies for highly customized procedures. The introduction of the HugoTM RAS System aimed to provide an alternative robotic platform that offers a more ergonomic and personalized working environment. Among its notable technical advantages are a more ergonomic trocar position, a larger working space for the bedside assistant, and cost-effectiveness for individual procedures.

  1. Does any Asia centre or hospital use the System? Any causes?

The main text was improved accordingly and a reference to Asia centres using the system was added as follow:

HugoTM RAS is currently being used in different hospitals in Asia which have found its main field of application in general surgery procedures, such as cholecystectomies, inguinal hernia repairs, lower anterior resections and gastric bypasses and in urologic surgeries such as prostatectomies. It was indeed the latter, the very first procedure carried out with Medtronic’s HugoTM RAS System in the Asia Pacific Region by Dr. Ragavan in 2021. Furthermore, the recent regulatory approval of the robotic system by Japan, the third largest robotic market worldwide, represents an outstanding expansion of minimally invasive surgeries in Asia, thus making the platform more accessible to a larger number of surgeons.

Ragavan, N., Bharathkumar, S., Chirravur, P., Sankaran, S., & Mottrie, A. (2022). Evaluation of Hugo RAS System in Major Urologic Surgery: Our Initial Experience. Journal of endourology, 36(8), 1029–1035. https://doi.org/10.1089/end.2022.0015

  1. A graphic study design is needed.

We thank the reviewer for the suggestion, a graphic study design was added.

Figure 1. Graphic study design showing the flow-chart of study selection.

  1. Tables can be used to demonstrate the applications among urology, gynaecology, colectomy, and general surgery.

We thank the reviewer for the suggestion, a table was added to demonstrate the applications among urology, gynaecology, and general surgery.

Comments on the Quality of English Language

The paper writing should be improved a lot.

We thank the Reviewer for the punctual comment. The whole manuscript has been revised by a native English mother tongue and improved accordingly.

For example, Line 87-89, Corrected sentence: The first case series of five robot-assisted radical prostatectomies (RARPs) ± lymph node dissection performed with the new HugoTM RAS system in Europe was described by C. A. Bravi et al [1].

The sentence was improved as suggested.

Lines 105-106, Corrected sentence: C. G. Alfano et al [2] retrospectively analyzed 15 consecutive patients who underwent RARP with the HugoTM RAS System from June to October 2021 in this robotic experience.

The sentence was improved as suggested.
